# IL-6 Inhibition as a Therapeutic Target in Aged Experimental Autoimmune Encephalomyelitis

**DOI:** 10.3390/ijms25126732

**Published:** 2024-06-19

**Authors:** María Dema, Herena Eixarch, Mireia Castillo, Xavier Montalban, Carmen Espejo

**Affiliations:** 1Servei de Neurologia, Centre d’Esclerosi Múltiple de Catalunya (Cemcat), Vall d’Hebron Institut de Recerca (VHIR), Hospital Universitari Vall d’Hebron, 08035 Barcelona, Spain; maria.dema@vhir.org (M.D.); herena.eixarch@vhir.org (H.E.); mcastillo@cem-cat.org (M.C.); xavier.montalban@cem-cat.org (X.M.); 2Universitat Autònoma de Barcelona, 08193 Bellaterra, Spain

**Keywords:** experimental autoimmune encephalomyelitis, IL-6, immunosenescence, ageing, innate immunity, multiple sclerosis

## Abstract

Multiple sclerosis (MS) onset at an advanced age is associated with a higher risk of developing progressive forms and a greater accumulation of disability for which there are currently no effective disease-modifying treatments. Immunosenescence is associated with the production of the senescence-associated secretory phenotype (SASP), with IL-6 being one of the most prominent cytokines. IL-6 is a determinant for the development of autoimmunity and neuroinflammation and is involved in the pathogenesis of MS. Herein, we aimed to preclinically test the therapeutic inhibition of IL-6 signaling in experimental autoimmune encephalomyelitis (EAE) as a potential age-specific treatment for elderly MS patients. Young and aged mice were immunized with myelin oligodendrocyte protein (MOG)_35–55_ and examined daily for neurological signs. Mice were randomized and treated with anti-IL-6 antibody. Inflammatory infiltration was evaluated in the spinal cord and the peripheral immune response was studied. The blockade of IL-6 signaling did not improve the clinical course of EAE in an aging context. However, IL-6 inhibition was associated with an increase in the peripheral immunosuppressive response as follows: a higher frequency of CD4 T cells producing IL-10, and increased frequency of inhibitory immune check points PD-1 and Tim-3 on CD4^+^ T cells and Lag-3 and Tim-3 on CD8^+^ T cells. Our results open the window to further studies aimed to adjust the anti-IL-6 treatment conditions to tailor an effective age-specific therapy for elderly MS patients.

## 1. Introduction

Aging of the immune system, which is called “immunosenescence”, is associated to the production of a senescence-associated secretory phenotype (SASP), characterized by the production of proinflammatory cytokines, chemokines, growth factors and extracellular matrix proteases that create a senescent microenvironment [1]. Among the components of the SASP, IL-6 is considered one of the most prominent cytokines. In fact, this innate immunity cytokine is overexpressed and secreted by senescent cells and strongly associates with chronic diseases and mortality in the elderly population [2]. IL-6 is a determinant for the development of autoimmunity and neuroinflammation, because it is involved in multiple sclerosis (MS) immunopathogenesis [3].

MS is a chronic, inflammatory, demyelinating and neurodegenerative disease of the central nervous system (CNS) of unknown etiology that commonly manifests between the ages of 20 and 35 years. However, MS patients that are diagnosed at older ages (late-onset MS) present an increased risk of developing progressive forms of the disease and greater accumulation of disability. Additionally, a high percentage of patients with a relapsing–remitting form of the disease gradually develop with time to secondary progressive MS, independently of disease duration [4]. Progressive MS patients have limited therapeutic options because there are currently no effective treatments available that can stop disability accumulation [5]. In addition, premature age-related changes of the immune system have been described in experimental autoimmune encephalomyelitis (EAE) models and MS patients [6]. MS is primarily mediated by adaptive immune responses, but it is widely accepted that progressive forms of the disease are mediated by compartmentalized innate immune responses in the CNS [7], supporting the fact that current available treatments that target the peripheral adaptive immune response are not effective in progressive patients.

Previous results from our group showed that aged mice presented increased severity of EAE disability together with a higher production of IL-6 in peripheral immune cells and in the CNS [8]. These results led us to conduct a preclinical study based on the inhibition of IL-6 signaling in EAE with an age equivalent to 50 years old in humans, when progressive forms of MS tend to appear, as a potential age-specific treatment for elderly MS patients. Although the clinical course of EAE was not improved in an aging context, IL-6 inhibition was associated with an increase in the peripheral immunosuppressive response. Our results open the field to tailor an effective age-specific therapy for MS patients who have developed progressive forms of MS at older ages.

## 2. Results

### 2.1. Blocking of IL-6 Signaling Did Not Ameliorate EAE Clinical Outcome in Aged Mice

In order to test the therapeutic effect of anti-IL-6 antibody in EAE with age, we treated mice every other day with anti-IL-6 antibody or IgG isotype once clinical signs were established. Young mice treated with anti-IL-6 antibody or IgG isotype were included to assess the specificity of the therapy with age. Anti-IL-6 antibody treatment did not ameliorate the clinical course of EAE in an aging context (Table 1).

### 2.2. EAE Clinical Parameters at 28 Days Postimmunization in Young and Aged Mice

Data represent two pooled independent experiments with nine mice per treatment and age for each experiment, accounting for a total of n = 18 (anti-IL-6 antibody) and n = 18 (IgG isotype) mice for young mice and n = 18 (anti-IL-6 antibody) and n = 18 (IgG isotype) for aged mice. Variables were analyzed using a two-way ANOVA test for maximum clinical score, motor coordination, clinical course (AUC) and weight loss (AUC), and statistical significance correction for multiple comparisons was performed with Bonferroni adjustment and the Log-rank test for time to score 3 and 4. Data are expressed as the mean ± SD. AUC is the area under the curve and sec is seconds.

We found no statistically significant differences in disability accumulation (Figure 1A), weight loss (Figure 1B), maximum clinical score nor motor coordination after the treatment. Moreover, both anti-IL-6 and IgG isotype–treated aged mice reached a mild tetraparesis (Figure 1C) and tetraparesis (Figure 1D) similarly in time. According to the clinical outcome, histopathological analysis showed no differences in the inflammatory infiltration after anti-IL-6 antibody treatment in aged mice (Figure 1E).

### 2.3. IL-6 Inhibition Enhances Peripheral Immunosuppressive Response in EAE Aged Mice

Next, we analyzed whether anti-IL-6 antibody treatment affects the peripheral immune response. We focused our interest on the study of cell populations that we had previously observed altered in aged EAE mice [8] and, at the same time, targeted by this cytokine. Our results showed an increase in the frequency of CD4^+^ T cells secreting IL-10 in anti-IL-6-treated aged mice (Table 2) and an age-specific decrease in helper T (Th)17/regulatory T (Treg) ratio in EAE after anti-IL-6 antibody treatment (Figure 2a), reflecting a predominance in the Treg over the Th17 response after IL-6 blockade. However, anti-IL-6 antibody treatment did not modify the Th1/Treg ratio (Figure 2b), the Th1/Th17 ratio (Figure 2c) nor the /Th2 ratio (Figure 2d) either in aged mice.

### 2.4. Th Responses and Expression of Immune Checkpoint Inhibitors in Te Cells at 28 dpi in Young and Aged Mice Treated with Anti-IL-6 Antibody or IgG Isotype

Data represent two pooled independent experiments with eight to nine mice per treatment and age for each experiment, accounting for a total of n = 18 (anti-IL-6 antibody) and n = 18 (IgG isotype) for young mice and n = 17 (anti-IL-6 antibody) and n = 17 (IgG isotype) for aged mice. Only mice that reached the endpoint were included in the analysis. Variables were analyzed using a GLIMMIX test and statistical significance correction for multiple comparisons was performed with Bonferroni adjustment. Data are expressed as the mean of the percentage ± SD. Statistical significance is in blue. LAG-3 is the lymphocyte activation gene 3, PD-1 is the programmed cell death protein 1, Te is the effector T, Th is the helper T, TIM-3 is the T cell immunoglobulin and mucin-domain containing-3 and Treg is the regulatory T.

Concerning the expression of the immune checkpoint inhibitors PD-1, LAG-3 and TIM-3, we found an age-specific increase in these inhibitory molecules in effector T (Te) cells after anti-IL-6 antibody treatment (Table 2). Specifically, we observed an increase in PD-1 (Figure 2e) and TIM-3 (Figure 2g) but not in LAG-3 (Figure 2f) within the CD4^+^ Te cell compartment, whereas LAG-3 (Figure 2i) and TIM-3 (Figure 2j) were found increased in the CD8^+^ Te cell compartment. In fact, no differences were found in the expression of PD-1 by CD8^+^ Te cells (Figure 2h). As with the Treg response, our results point out that anti-IL-6 antibody treatment enhances the expression of immune checkpoint inhibitors in aged Te cells in EAE, both in CD4 and CD8 compartments, but does not modify the immunoregulatory properties of Te in young mice. On the other hand, anti-IL-6 treatment did not affect the cytotoxic capacity of CD8^+^ T, NKT and NK cells, because we found no differences in the production of granzyme B or perforin in either cell population after anti-IL-6 antibody treatment with age (Table 3).

### 2.5. CD8+ T, NKT and NK Cell Cytotoxic Capacity at 28 dpi in Young and Aged Mice Treated with IgG Isotype or Anti-IL-6 Antibody

Data represent two pooled independent experiments with eight to nine mice per treatment and age for each experiment, accounting for a total of n = 18 (anti-IL-6 antibody) and n = 18 (IgG isotype) for young mice and n = 17 (anti-IL-6 antibody) and n = 17 (IgG isotype) for aged mice. Only mice that reached the endpoint were included in the analysis. Variables were analyzed using a GLIMMIX test and statistical significance correction for multiple comparisons was performed with Bonferroni adjustment. Data are expressed as the mean of the percentage ± SD. Statistical significance is in blue. GzmB is granzyme B, NK is natural killer, NKT is natural killer T and Prf is perforin.

## 3. Discussion

The number of elderly MS patients is growing [9]. The fact that MS differs between elderly and young patients [4] suggests that the underlying immune mechanisms of the disease are different in both populations and that immunosenescence may have a relevant role in disease progression. Aging of the immune system is associated with the production of a bioactive secretome known as the SASP. IL-6 is considered one of the most prominent cytokines of the SASP and a determinant for the development of autoimmunity and neuroinflammation, because IL-6 is involved in MS immunopathogenesis. Indeed, we have previously described higher IL-6 production in the peripheral immune cells and increased expression in the CNS in the context of EAE with aging [8].

IL-6 is principally produced by lymphocyte, myeloid and endothelial cells and is a major cytokine in the CNS produced by neurons and glial cells. The involvement and overproduction of IL-6 has been reported in many autoimmune diseases, including MS [3]. IL-6 knockout mice show impaired lymphocyte proliferation, lower autoreactivity to myelin oligodendrocyte glycoprotein (MOG) and no inflammatory infiltration into the CNS, probably as a result of the impaired lymphocyte–endothelial interaction in the blood–brain barrier. The lack of IL-6 translates into a resistance to EAE, emphasizing the immunopathological role of this cytokine in MS [10,11]. Peripheral immune cells secreting this cytokine have a relevant role in the disease, such as dendritic cells (DCs), Treg cells and B cells. DC-specific IL-6 deficiency leads to a higher resistance to EAE, because naïve T cells cannot differentiate into autoreactive Th17 cells. Treg cell–specific IL-6 depletion results in a milder clinical course of the disease, although it does not affect the susceptibility to develop EAE [12]. Similarly, B cell–specific IL-6–deficient mice develop an attenuated form of EAE. Considering that autoantibody levels were unaffected, this specific depletion indicates that IL-6 production is one of the main nonantibody-mediated mechanisms of pathogenic B cells involved in the development of EAE [13,14]. However, the conditional depletion of IL-6 in the CNS is not as effective as in peripheral immune cells, because depletion of IL-6 in astrocytes and microglia can eventually ameliorate but not prevent EAE [15]. Preventive treatment with anti-IL-6 or anti-IL-6 receptor (IL-6R) antibodies improves the clinical course of young mice with EAE. This protective effect is believed to be mediated via the suppression of MOG-specific Th17 response and reduced infiltration of these T cells and macrophages into the CNS [16,17]. However, the same treatment was not effective when it was performed in a therapeutic approach, a possible explanation being that the commitment to Th17 cells had already occurred [17]. Nevertheless, when the same anti-IL-6R antibody was given orally and cells were adoptively transferred into active EAE recipient mice, CNS inflammation and Th1 responses were reduced, and Th2 responses were found increased, resulting in an amelioration of EAE [18]. Focusing on our results, a blockade of IL-6 did not ameliorate the clinical outcome in an age-specific manner, indeed neither in EAE young mice as previously described [17]. Instead, we observed changes in the balance between Th17 and Treg responses, which probably corresponds to the inhibition of newly differentiating Th17 cells and promotion of the differentiation of Treg cells when IL-6 is inhibited. These changes in the peripheral immune system are not sufficient to ameliorate the clinical outcome of EAE, suggesting that anti-IL-6 at our dosage and frequency administered is not able to inhibit the proliferation of already committed pathogenic Th17 cells in a context of age. We also observed an age-specific increase in the expression of immune checkpoint inhibitors after anti-IL-6 antibody treatment, because PD-1, LAG-3 and TIM-3 were enhanced in CD4^+^ and CD8^+^ Te cells. In fact, there is evidence sustaining the increase in the suppressive response after blocking this cytokine. Highly suppressive Treg cells are able to inhibit the proliferation of effector CD4^+^ T cells after anti-IL-6R treatment [19]. Moreover, the inhibition of IL-6 signaling is known to shift the balance toward immune regulation, by restoring the number and function of peripheral Treg cells and dampening activated and effector CD4^+^ T cell function [20,21,22,23]. Regarding the immune checkpoint inhibitors, there is an increase in TIM-3 in peripheral lymphocytes from arthritis rheumatoid patients treated with anti-IL-6R antibody [24]. Although IL-6 is a potent immune-mediator in the context of neuroinflammation, we found no differences in the inflammatory environment in the CNS of those aged EAE mice treated with anti-IL-6. Despite the increase in the peripheral suppressive response after anti-IL-6 antibody treatment, this response seems to be insufficient in order to revert the clinical outcome observed in aged mice. This fact could be due to an inadequate dose [17], administration route [18] or days of the treatment [16,17], because some of these studies used higher quantities of the antibody, different routes and less days of administration compared to our experimental design. Because the current work was designed as an exploratory study, we did not test different doses, frequency of treatment and routes of administration. Moreover, sex differences were not studied because female mice are preferred in EAE studies due to their higher susceptibility to the disease, the significant impact of hormonal differences on immune responses and the greater clinical relevance of findings to human MS, which predominantly affects women. Regarding the route of administration, we chose intraperitoneal (i.p.) administration because it is a commonly used route in exploratory studies like ours, facilitating effective absorption of compounds into the bloodstream. Additionally, i.p. administration is less stressful and harmful than intravenous when frequent treatment is required [25].

In conclusion, the present study points out that the clinical outcome of EAE cannot be reverted by blocking IL-6 signaling in an age-specific manner. Nonetheless, anti-IL-6 therapy is able to increase the peripheral immunosuppressive response in an aging context, probably leading to the slight amelioration observed in the clinical course of aged mice. Our studies open the window to anti-IL-6–based therapies for the development of an age-specific effective treatment for elderly MS patients.

## 4. Materials and Methods

### 4.1. Mice

C57BL/6JRccHsd 8-week-old (20 years old in humans) and 40-week-old (50 years old in humans) [26] female mice (Envigo, Horst, The Netherlands) were housed under standard light and climate-controlled conditions and standard chow and water were provided ad libitum. All experiments were performed in strict accordance with European Union (Directive 2010/63/EU) and Spanish regulations (Real Decreto 53/2013; Generalitat de Catalunya Decret 214/97). The Ethics Committee on Animal Experimentation of the Vall d’Hebron Research Institute approved all procedures described in this study (protocol number: 67/18 CEEA; CEA-OH/10683/1). All data presented are in accordance with the guidelines suggested for EAE publications [27] and the Animal Research: Reporting of In Vivo Experiments (ARRIVE) guidelines for animal research [28].

### 4.2. EAE Induction and Clinical Follow-Up

Anesthetized mice were immunized by subcutaneous injections of 100 µL phosphate buffered saline (PBS) 1× containing 100 µg of rat peptide 35–55 of MOG (Proteomics Section, Universitat Pompeu Fabra, Barcelona, Spain) emulsified in 100 µL of complete Freund’s adjuvant (incomplete Freund’s adjuvant (IFA, F5506, Merck, Darmstadt, Germany) containing 4 mg/mL Mycobacterium tuberculosis H37RA (231141, BD, Franklin Lakes, NJ, USA)). At 0- and 2-days postimmunization (dpi), the mice were intravenously injected with 250 ng of pertussis toxin (P7208, Merck). The mice were weighed and examined daily for neurological signs in a blinded manner using the following criteria: 0, no clinical signs; 0.5, partial paresis of tail; 1, paralysis of whole tail; 2, mild paraparesis of one or both hind limbs; 2.5, severe paraparesis or paraplegia of hind limbs; 3, mild tetraparesis (mild in hind limbs); 3.5, moderate tetraparesis (moderate in hind limbs); 4, tetraparesis (severe in hind limbs); 4.5, severe tetraparesis; 5, tetraplegia; and 6, death [29]. If weight loss was greater than 15%, the mice received subcutaneous administration of 0.5 mL of 10% glucose and endpoint criteria was defined as a weight loss greater than 30% or a clinical score of 5. Regarding the clinical parameters, the overall clinical score was calculated as the area under the curve (AUC) of the daily clinical score throughout the experiment. The overall accumulated weight loss was calculated as the AUC of the curve representing the daily percentage of weight loss in respect to the initial weight on the day of immunization. The maximum clinical score was defined as the maximum punctuation obtained during the clinical follow-up. The mice were euthanized at the end of the clinical follow-up (28 dpi) using carbon dioxide >70% and the spinal cord and spleen were collected.

### 4.3. Experimental Treatment

Two independent experiments were performed. Young and aged mice that reached a clinical score equal to or greater than 1 (between 12 and 18 dpi) were randomly allocated into clinically equivalent experimental groups, based on the clinical onset, days of disease and accumulated score (Table 4). Intraperitoneal administration of 15 mg/kg of InVivoMAb AntiMouse IL-6 (Rat, IgG1 kappa, clone MP5-20F3, BE0046, Bio X Cell, Lebanon, NH, USA) or InVivoMab Anti-HRP Isotype Control (Rat, IgG1 kappa, clone HRPN, BE0088, Bio X Cell) diluted in InVivoPure pH 7.0 Dilution Buffer (IP0070, Bio X Cell) was performed every other day from the day of randomization until the end of the experiment. See Figure 3 for details about the experimental design. The dose and frequency of administration of the anti-IL-6 antibody was selected taking into account the regimen used with blocking antibodies in other studies [30,31,32].

### 4.4. Motor Function Assessment

Motor performance was evaluated thrice for each mouse at the end of the preclinical study using a Rotarod (Ugo Basile, Gemonio, Italy), which was set to accelerate from a speed of 4 to 40 rotations per minute (rpm) in a trial of 300 s. After two training trials at a constant speed, the mice were placed on the rotating cylinder and the time walking without falling was recorded.

### 4.5. Histopathological Analysis

Coronal sections of lower spinal cord embedded in paraffin were deparaffinized, rehydrated and hematoxylin and eosin (H&E) staining was performed. Images were acquired using a NanoZoomer slide scanner and NDP.view2 visualization software version U12388-01 (Hamamatsu Photonics, Hamamatsu, Japan). For each mouse, two mosaic tiles of the thoracic spinal cord separated by 300 µm were analyzed. The percentage of total area with inflammatory infiltrates relative to the total white matter (WM) area was quantified using ImageJ software (Wayne Rasband, NIH, Bethesda, MD, USA). Spinal cords from one experiment were analyzed and those with missing areas in the WM were excluded from the analysis.

### 4.6. Peripheral Immune Response Analysis

Spleens were removed and maintained in noncomplete RPMI (Biowest, Bradenton, FL 34211, USA) at 4 °C overnight (ON). Splenocytes were isolated by mechanically grinding spleens through a 70 µm nylon cell strainer (Cell strainer 70 µm nylon, FalconR Corning, New York, NY, USA) using X-VIVO^TM^ 15 medium (BE-02-060F, Lonza, Basel, Switzerland) supplemented with 10 mM Hepes (H0887, Merck) and 6 µM 2-β-mercaptoethanol (M3148, Merck). Subsequently, a hypotonic lysis was performed using sterile water to eliminate the corresponding erythroid component from the cell suspension, and immediately buffered with PBS containing 1% fetal bovine serum (FBS; Biowest). Clumps were eliminated through a second 70 µm nylon cell strainer. Isolated splenocytes were stained using the corresponding combination of fluorochrome-conjugated antibodies (Table 5). A total of 5 panels were designed to analyze the immune cell populations of interest as follows: expression of immune check-point molecules in T cells, proinflammatory cytokine-producing CD4 T cells, anti-inflammatory cytokine-producing CD4^+^ T cells, regulatory T (Treg) cells and cytotoxicity of CD8^+^ T, NKT and NK cells. For the analysis of proinflammatory and anti-inflammatory cytokine-producing T cells, ex vivo stimulation of 2 × 10^6^ splenocytes was performed with 50 ng/mL phorbol 12-myristate 13-acetate (PMA, P1585, Merck) and 1 µg/mL ionomycin (I0634, Merck) in the presence of 5 µL of GolgiPlug (GP, 555029, BD) and 1.33 µL of GolgiStop (GS, 554724, BD) every 2 × 10^6^ cells during 6 h at 37 °C, 5% CO_2_. For analysis of cytotoxicity on CD8^+^ T, NKT and NK cells, ex vivo stimulation of 2 × 10^5^ splenocytes was performed with 50 ng/mL Recombinant Mouse IL-2 Protein (402-ML, Bio-Techne, Minneapolis, MN, USA) and 200 ng/mL Recombinant Murine IL-15 (210-15, Thermo Fisher Scientific, Waltham, MA, USA) for 72 h, 6 h at 37 °C, 5% CO_2_, and the last 6 h cells were cultured in the presence of 5 µL of GP and 1.33 µL of GS every 2 × 10^6^ cells [9]. Before the staining, cells were washed with PBS 1× and incubated with the corresponding fixable viability dye (see Table 5) for 15 min at room temperature (RT) to exclude dead cells. Then, cells were incubated RT for 10 min with 0.5 µg of rat antimouse CD16/32 antibody (553142, BD) every 10^6^ cells, in order to block Fc receptors of immune cells and avoid unspecific staining. Surface staining was performed at 4 °C for 30 min. Nuclear FoxP3 staining was performed with the FoxP3/Transcription Factor Staining Buffer Set (00-5523-00, Thermo Fisher Scientific), and incubation of anti-FoxP3 was performed at RT for 30 min. Intracellular staining of IFN-γ (Th1 response), IL-17 (Th17 response), IL-10 (Treg response), IL-4 (Th2 response), perforin and granzyme B was performed using the Cytofix/Cytoperm kit (554714, BD) followed by a 30 min incubation with intracellular antibodies at 4 °C. Samples were acquired in a CytoFLEX flow cytometer and data were analyzed with CytExpert 2.4 software (Beckman Coulter, Brea, CA, USA).

### 4.7. Statistical Analysis

All the analyses were performed in a blinded manner and only considering mice that reached the endpoint: one aged EAE mice treated with the isotype and another treated with the anti-IL-6 antibody reached a clinical score of 6, so they were excluded from the analysis. Statistical details and data representation for each study are described in the corresponding figure legend. EAE time to reach score 3 and 4 were analyzed using the Log-rank test. If two experimental groups were compared in an independent experiment, variables were analyzed using a *t*-test. If two or more experimental groups were compared in several independent experiments, variables were analyzed using a two-way ANOVA test or using a GLIMMIX generalized linear model and statistical significance correction for multiple comparisons was performed using Bonferroni adjustment. Data estimation was performed using the least-squares means method assuming a lognormal distribution, considering age and treatment as dependent variables and experiment as an independent variable. Data were expressed as the mean ± standard deviation (SD), except for the clinical score, which was represented as the mean ± standard error of the mean (SEM). Statistical significance was set at an adjusted *p*-value of <0.05. Statistical analyses were performed using SAS© 9.4 (SAS Institute Inc., Cary, NC, USA) and GraphPad Prism 8.0 (GraphPad, La Jolla, CA, USA).

## Figures and Tables

**Figure 1 ijms-25-06732-f001:**
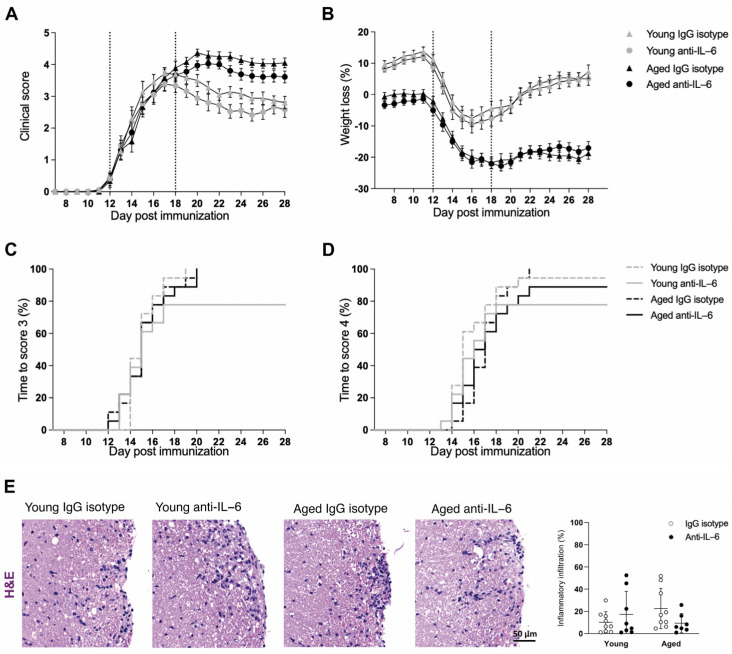
Treatment with anti-IL-6 did not ameliorate the clinical course nor reduce CNS inflammatory infiltration in aged EAE. (**A**) Clinical course, (**B**) weight loss, (**C**) time to reach a score of 3 (mild tetraparesis) and (**D**) time to reach a score of 4 (tetraparesis) in young and aged mice treated with anti-IL-6 antibody or IgG isotype. Data represent two pooled independent experiments with nine mice per treatment and age for each experiment, accounting for a total of n = 18 (IgG isotype) and n = 18 (anti-IL-6 antibody) for young mice, and n = 18 (IgG isotype) and n = 18 (anti-IL-6 antibody) for aged mice. Dotted lines indicate the treatment initiation period. (**E**) Quantification of inflammatory infiltration in the white matter (WM) spinal cord in young and aged mice treated with Ig isotype or anti-IL-6 antibody. Data represent an individual experiment with n = 9 (IgG isotype) and n = 8 (anti-IL-6 antibody) for young mice and n = 9 (IgG isotype) and n = 7 (anti-IL-6 antibody) for aged mice. Only mice that reached the endpoint were included in the analysis. Variables were analyzed using a two-way ANOVA test and statistical significance correction for multiple comparisons was performed with Bonferroni adjustment in (**A**,**B**), Log-rank test in (**C**,**D**) and *t*-test in (**E**). Data are expressed as the mean ± SEM in (**A**,**B**), as survival curves in (**C**,**D**) and as mean ± SD in (**E**). AUC: area under the curve; H&E: hematoxylin and eosin; WM: white matter.

**Figure 2 ijms-25-06732-f002:**
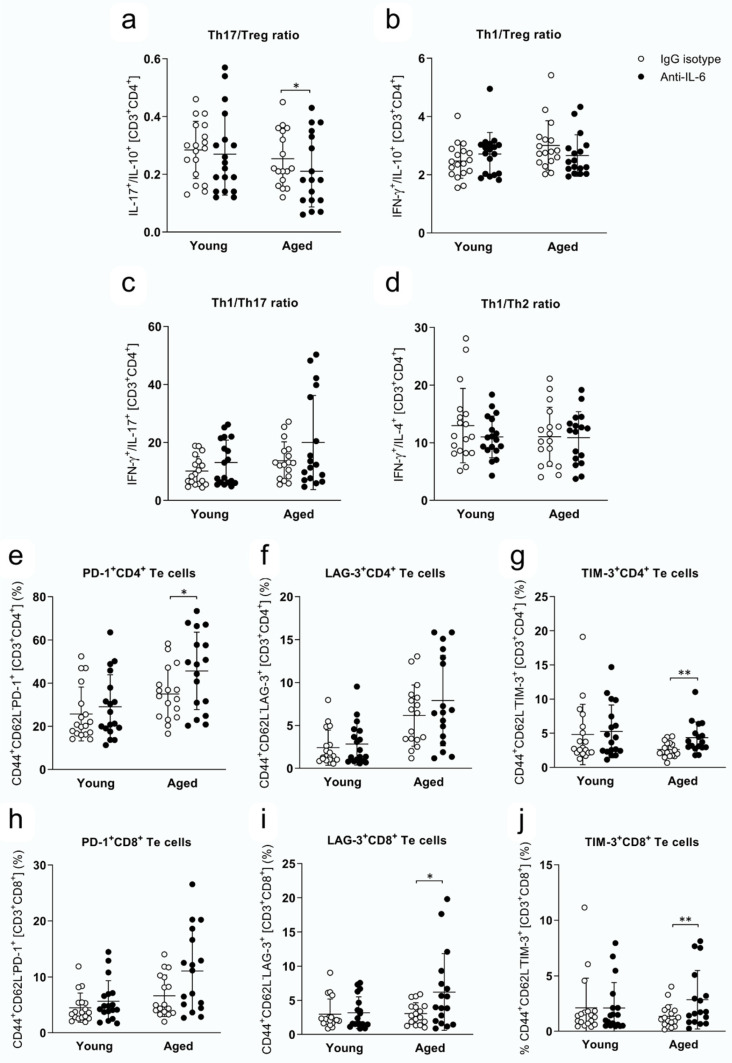
Anti-IL-6 antibody treatment increased peripheral immunosuppressive response in aged EAE. (**a**) Th1/Th17 ratio, (**b**) Th1/Treg ratio, (**c**) Th17/Treg ratio, (**d**) Th1/Th2 ratio, (**e**) PD-1^+^CD4^+^, (**f**) LAG-3^+^CD4^+^, (**g**) TIM-3^+^CD4^+^, (**h**) PD-1^+^CD8^+^, (**i**) LAG-3^+^CD8^+^, (**j**) TIM-3^+^CD8^+^ T effector cells at 28 days postimmunization in young and aged mice treated with IgG isotype or antiIL-6 antibody. Data represent two pooled independent experiments with eight to nine mice per treatment and age for each experiment, accounting for a total of n = 18 (IgG isotype) and n = 18 (anti-IL-6 antibody) for young mice and n = 17 (IgG isotype) and n = 17 (anti-IL-6 antibody) for aged mice. Only mice that reached the endpoint were included in the analysis. Variables were analyzed using a GLIMMIX test and statistical significance correction for multiple comparisons was performed with Bonferroni adjustment. Data are expressed as the ratio of cell frequencies in (**a**–**d**) and as mean ± SD in (**e**–**j**). * *p* < 0.05; ** *p* < 0.01. Te: effector T; Th: helper T; Treg: regulatory T.

**Figure 3 ijms-25-06732-f003:**
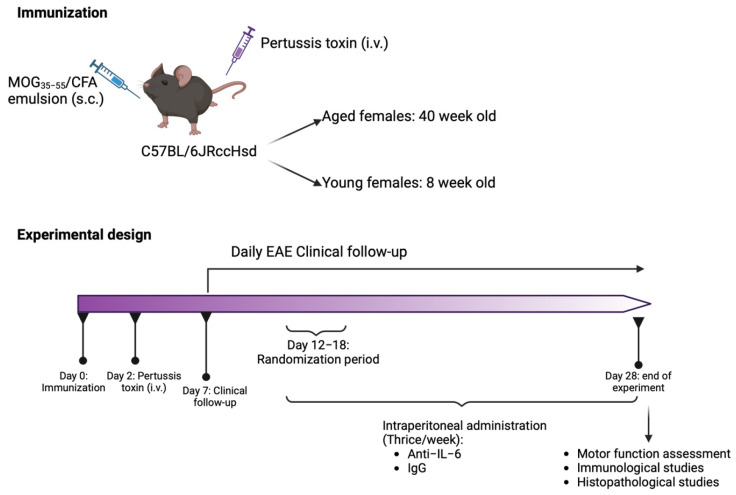
Scheme of the immunization conditions and experimental design of this study. Created with BioRender.com, accessed on 10 June 2024.

**Table 1 ijms-25-06732-t001:** Treatment with anti-IL-6 antibody does not modify EAE clinical parameters.

Parameter	Anti-IL-6	IgG Isotype	*p*-Value
Maximum clinical score	young: 3.6 ± 0.8	young: 4.1 ± 0.3	0.1016
	aged: 4.2 ± 0.6	aged: 4.4 ± 0.5	1.0000
Motor coordination (sec)	young: 26.9 ± 18.5	young: 19.0 ± 8.6	0.8482
	aged: 8.0 ± 7.1	aged: 6.4 ± 5.3	1.0000
Clinical score (AUC)	young: 42.3 ± 13.4	young: 47.9 ± 10.1	0.5078
	aged: 52.6 ± 11.6	aged: 56.0 ± 9.3	1.0000
Weight loss (AUC)	young: 38.6 ± 135.7	young: 58.7 ± 141.7	1.0000
	aged: −301.7 ± 114.2	aged: −290.9 ± 124.2	1.0000
Time to score 3	young: 14/18 (77.78%)	young: 18/18 (100%)	0.2381
	aged: 18/18 (100%)	aged: 18/18 (100%)	0.8146
Time to score 4	young: 14/18 (77.78%)	young: 17/18 (94.44%)	0.2913
	aged: 16/18 (88.89%)	aged: 18/18 (100%)	0.5464

**Table 2 ijms-25-06732-t002:** Age-related changes in Th responses and expression of immune checkpoint inhibitors in T effector cells in EAE after anti-IL-6 antibody treatment.

Population	Anti-IL-6	IgG Isotype	*p*-Value
Th1 response	young: 16.05 ± 6.82	young: 15.13 ± 7.27	0.9448
	aged: 24.45 ± 8.66	aged: 19.50 ± 6.52	0.2272
Th2 response	young: 1.71 ± 1.19	young: 1.47 ± 1.30	0.2769
	aged: 2.63 ± 1.48	aged: 2.01 ± 0.80	0.2900
Th17 response	young: 1.51 ± 0.67	young: 1.63 ± 0.58	0.7276
	aged: 1.74 ± 0.79	aged: 1.61 ± 0.59	0.7399
Treg response	young: 6.08 ± 2.32	young: 6.26 ± 2.88	1.0000
	aged: 9.50 ± 3.52	aged: 6.76 ± 2.55	0.0387
Th1/Th17 ratio	young: 13.06 ± 7.72	young: 10.12 ± 4.97	0.2184
	aged: 19.22 ± 16.05	aged: 13.69 ± 6.51	0.2875
Th1/Treg ratio	young: 2.71 ± 0.74	young: 2.47 ± 0.60	0.7924
	aged: 2.66 ± 0.71	aged: 3.00 ± 0.85	1.0000
Th17/Treg ratio	young: 0.27 ± 0.14	young: 0.28 ± 0.10	0.7686
	aged: 0.21 ± 0.12	aged: 0.25 ± 0.10	0.0387
Th1/Th2 ratio	young: 11.01 ± 3.62	young: 12.97 ± 6.45	0.3342
	aged: 10.88 ± 4.51	aged: 11.03 ± 5.12	0.1290
PD-1^+^CD4^+^ Te cells	young: 29.03 ± 14.81	young: 25.68 ± 12.47	0.6461
	aged: 45.64 ± 17.93	aged: 34.96 ± 12.13	0.0466
LAG-3^+^CD4^+^ Te cells	young: 2.85 ± 2.45	young: 2.41 ± 2.05	1.0000
	aged: 7.90 ± 5.27	aged: 6.15 ± 3.56	0.7331
TIM-3^+^CD4^+^ Te cells	young: 5.27 ± 3.87	young: 4.82 ± 4.42	0.9006
	aged: 4.34 ± 2.31	aged: 2.61 ± 1.12	0.0052
PD-1^+^CD8^+^ Te cells	young: 5.65 ± 3.68	young: 4.49 ± 2.63	0.6900
	aged: 11.10 ± 7.27	aged: 6.64 ± 3.72	0.0563
LAG-3^+^CD8^+^ Te cells	young: 3.18 ± 2.33	young: 2.97 ± 2.24	1.0000
	aged: 6.22 ± 5.62	aged: 3.06 ± 1.60	0.0252
TIM-3^+^CD8^+^ Te cells	young: 2.10 ± 2.30	young: 2.12 ± 2.68	1.0000
	aged: 2.85 ± 2.64	aged: 1.36 ± 1.05	0.0294

*p*-values highlighted in blue indicate statistical significance.

**Table 3 ijms-25-06732-t003:** Age-related changes in CD8^+^ T, NKT and NK cell cytotoxic capacity in EAE after anti-IL-6 antibody treatment.

Population	Anti-IL-6	IgG Isotype	*p*-Value
Prf^+^CD8^+^ T cells	young: 70.97 ± 12.47	young: 72.24 ± 13.72	1.0000
	aged: 65.68 ± 19.26	aged: 70.69 ± 16.75	0.3293
GzmB^+^CD8^+^ T cells	young: 41.22 ± 12.07	young: 48.09 ± 8.95	0.1504
	aged: 42.72 ± 15.00	aged: 47.47 ± 11.84	0.3689
Prf^+^ NKT cells	young: 73.63 ± 14.96	young: 75.42 ± 15.28	1.0000
	aged: 78.94 ± 10.34	aged: 82.17 ± 11.98	0.9413
GzmB^+^ NKT cells	young: 46.36 ± 13.56	young: 56.43 ± 14.98	0.0129
	aged: 49.50 ± 15.10	aged: 54.26 ± 16.14	0.4372
Prf^+^ NK cells	young: 96.64 ± 2.92	young: 97.72 ± 1.39	0.6017
	aged: 93.66 ± 4.80	aged: 95.17 ± 3.67	0.2974
GzmB^+^ NK cells	young: 86.35 ± 8.66	young: 90.36 ± 6.57	0.4755
	aged: 76.34 ± 16.38	aged: 76.30 ± 15.11	1.0000

*p*-values highlighted in blue indicate statistical significance.

**Table 4 ijms-25-06732-t004:** Randomization criteria for anti-IL-6 treatment. Young I and Aged I refer to clinical parameters at the time of randomization in experiment 1. Young II and Aged II refer to clinical parameters at the time of randomization in experiment 2.

Age	Treatment	Clinical Onset	Days of Disease	Accumulated Score
Young I	IgG isotype (n = 9)	12.9 ± 1.6	2.0 ± 0.5	3.4 ± 2.0
	Anti-IL-6 (n = 9)	12.9 ± 1.6	2.0 ± 0.5	3.3 ± 2.3
Aged I	IgG isotype (n = 9)	14.3 ± 1.8	1.4 ± 0.5	2.4 ± 0.7
	Anti-IL-6 (n = 9)	14.1 ± 2.1	1.7 ± 0.7	2.6 ± 0.8
Young II	IgG isotype (n = 9)	13.1 ± 1.5	1.8 ± 0.4	2.2 ± 0.9
	Anti-IL-6 (n = 9)	12.9 ± 1.5	1.9 ± 0.6	2.3 ± 0.9
Aged II	IgG isotype (n = 9)	13.3 ± 1.1	1.3 ± 0.5	2.6 ± 0.5
	Anti-IL-6 (n = 9)	13.2 ± 1.0	1.4 ± 0.5	2.8 ± 1.1

**Table 5 ijms-25-06732-t005:** Antibodies used in flow cytometry studies.

Panel	Antibody	Fluorochrome	Clone	Reference	Manufacturer	Isotype
Proinflammatory cytokine-producing T cells	IFN-γ	BV421	XMG1.2	563376	BD	Rat IgG1k
	FVS	BV510	-	564406	BD	-
	CD8	BV605	53-6.7	563152	BD	Rat IgG2ak
	CD3	FITC	145-2C11	561827	BD	Hamster IgG1k
	CD4	PerCP-eF710	RM4-5	46-0042	Thermo Fisher Scientific	Rat IgG2ak
	IL-17a	PE	TC11-18H10	561020	BD	Rat IgG1k
	CD69	APC	H1.2F3	560689	BD	Hamster IgG1λ1
Anti-inflammatory cytokine-producing T cells	IL-10	BV421	JES5-16E3	563276	BD	Rat IgG2bk
	FVS	BV510	-	564406	BD	-
	CD8	BV605	53-6.7	563152	BD	Rat IgG2ak
	CD3	FITC	145-2C11	561827	BD	Hamster IgG1k
	CD4	PerCP-eF710	RM4-5	46-0042	Thermo Fisher Scientific	Rat IgG2ak
	IL-4	APC	11B11	554436	BD	Rat IgG1k
Treg cells	FVS	BV510	-	564406	BD	-
	CD8	BV605	53-6.7	563152	BD	Rat IgG2ak
	CD3	FITC	145-2C11	561827	BD	Hamster IgG1k
	CD25	PE	PC61	12-0251	BioLegend	Rat IgG1λ
	CD39	PE-Cy7	24DMS1	25-0391	Thermo Fisher Scientific	Rat IgG2bk
	FoxP3	APC	FJK-16s	17-5773	BD	Rat IgG2ak
	CD4	APC-H7	GK1.5	560181	BD	Rat IgG2ak
CD8^+^ T, NKT and NK cell cytotoxicity	Granzyme B	eF-450	NGZB	48-8898	Thermo Fisher Scientific	Rat IgG2ak
	CD8	BV605	53-6.7	563152	BD	Rat IgG2ak
	CD3	FITC	145-2C11	561827	BD	Hamster IgG1k
	CD4	PerCP-eF710	RM4-5	46-0042	Thermo Fisher Scientific	Rat IgG2ak
	Perforin	PE	S16009A	154306	BioLegend	Rat IgG2ak
	NK1.1	APC	PK136	550627	BioLegend	Mouse IgG2ak
	FVS	APC-700	-	564997	BD	-
Immune checkpoint inhibitors	CD279	BV421	MIH4	564323	BD	Rat IgG2ak
	CD8	BV510	53-6.7	563068	BD	Rat IgG2ak
	CD62L	BV605	MEL-14	563252	BD	Rat IgG2ak
	CD3	FITC	145-2C11	561827	BD	Hamster IgG1k
	CD44	PerCP-Cy5.5	IM7	560570	BD	Rat IgG2bk
	CD366	PE-Cy7	RMT3-23	25-5870	BD	Rat IgG2ak
	CD223	APC	C9B7W	562346	BD	Rat IgG2ak
	FVS	APC-700	-	564997	BD	-
	CD4	APC-H7	GK1.5	560181	BD	Rat IgG2ak

Antibodies highlighted in blue are intracellular markers.

## Data Availability

Further information and requests for resources and reagents should be directed to and will be fulfilled by the lead contact, Carmen Espejo (carmen.espejo@vhir.org).

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
