# Peer review of "IL-6 Inhibition as a Therapeutic Target in Aged Experimental Autoimmune Encephalomyelitis"

_ijms, 2024, doi:10.3390/ijms25126732_

Round 1

Reviewer 1 Report

Comments and Suggestions for Authors

I appreciate the opportunity to review the manuscript entitled "IL-6 inhibition as a therapeutic target in aged experimental autoimmune encephalomyelitis". The overall study seemed well-designed. 

The authors should take note of the major and minor remarks listed below to improve the manuscript:

Major comments:

1. My major concern is on what basis the dose and frquency were decided. Authors have mentioned that this is based on other studies; however, they have not provided any reference to this. Moreover, reference number 8 is not published yet. Therefore, it is very difficult to review the appropraieness of dosing and frequency. 

2. In table 4, mention of Young II and Aged II is unclear, as there are two groups in this study: young and age. If this refers to FIgure 1, please modify the figure accordingly for clarity.

3.  I am curious to know why only PD-1, LAG-3, and TIM-3 were used as ICIs. There are many others (please refer to https://doi.org/10.1186/s12964-023-01289-9). Please discuss. 

4. Why was the intraperitoneal route used for antibody administration? Please discuss in the manuscript.

5. Please provide limitations of the study.

Minor comments:

1. Please mention time and temperature in section 4.6. Also, kindly provide citation or brie method for splenocyte isolation

2. Please provide a figure/scheme for EAE induction and treatment.

Comments on the Quality of English Language

The article is simple and well written. There are few typographical and grammatical mistakes that need to be addressed. For example, line no. 66: "established" and not "stablished."

Author Response

Thank you for taking the time to review our manuscript entitled. We greatly appreciate your constructive feedback. Your comments and suggestions have been essential in enhancing the quality and clarity of our study.

Below, we have attempted to address your questions and comments point by point. We remain at your disposal for any further comments.

Major comments:

1. We thank the reviewer for the comment, not providing references for dose and frequency choice was a mistake we did not realise at the time of submitting the manuscript. We have added three references concerning dosage of anti-IL6 treatment (line 293). As the reviewer can see, the dosage and frequency are not homogeneous among different studies, however we decide to use an average dose, and a frequency of every other day administration in order to avoid a putative loss of biological blockade of IL-6.

We cite the reference 8 as the guide for authors of the journal indicates, as submitted to a journal. In our opinion it is worth mentioning that IL-6 was selected as a possible target molecule based on previous work done in the lab, even though it has not already been published.

2. We have added additional information on the table captions to specify that data correspond to clinical parameters at the time of randomization of two independent experiments. For the sake of clarity, we also detail in Figure legends that represented data come from two pooled independent experiments with n=9 for each experimental group and age.

Additionally, we provide (see below) the reviewer a table with the clinical parameters at the time of randomization for both experiments (as it is in Figure 1), and separate representation of clinical parameters of both experiments separately (as it is in Table 4 in the manuscript).

Although we prefer to maintain the current representation of data, we can change Figure 1 or Table 4 if the reviewer thinks that changes we have done are not clear enough.

Age

Treatment

Clinical onset

Days of disease

Accumulated score

Young

IgG isotype (n=18)

13.0 ± 1.5

1.9 ± 0.5

2.8 ± 1.6

Anti-IL-6 (n=18)

12.9 ± 1.5

1.9 ± 0.5

2.8 ± 1.7

Aged

IgG isotype (n=18)

13.8 ± 1.5

1.4 ± 0.5

2.5 ± 0.6

Anti-IL-6 (n=18)

13.7 ± 1.6

1.6 ± 0.6

2.7 ± 0.9

3. We greatly appreciate the opportunity to review the paper provided by the reviewer. The paper offers an extensive analysis of immune checkpoint inhibitors (ICIs) in the context of multiple sclerosis, making it highly valuable to our research. As the reviewer correctly points out, there is a wide array of ICIs that could be investigated. However, we have specifically focused on inhibitory ICIs, selecting three that are particularly relevant to multiple sclerosis, as highlighted in the review and from our own unpublished data in other therapeutic approaches in the EAE model.

We concur that expanding the range of ICIs under study would yield a greater breadth of information. Nonetheless, we trust that the reviewer will agree that focusing initially on PD-1, Tim-3, and Lag-3 is a worthwhile approach. These particular ICIs are highly relevant to the context of multiple sclerosis and provide a solid foundation for our research. We envisage that subsequent studies will indeed encompass a broader spectrum of ICIs, building on the insights gained from this initial investigation.

We acknowledge that this aspect was not discussed in the manuscript. However, should the reviewer deem it necessary, we are fully prepared to address and elaborate on this point in a revised version. We are committed to ensuring that our manuscript meets the highest standards of comprehensiveness and clarity.

4. We chose the i.p. administration route since it is the most commonly used for initial and proof-of concept studies, such as ours. I.p. administration usually results in a good absorption of compounds into the systemic circulation which facilitates their arrival to secondary lymphatic organs. In addition, i.p. administration allows serial administrations for long periods of time, while i.v. administration every other day is much more stressful for mice and animals are at higher risk of suffering injuries and wounds in the tail. For all these reasons we considered i.p. as the appropriate route of administration for our experiments. Further justification of the usage of the i.p. administration in initial experimental animal studies is extensively reviewed in Al Shoyaib et al, Pharm Res (2020); 37:12 DOI: 10.1007/s11095-019-2745-x.

We have added the reference and discussion in the manuscript.

5. A paragraph with limitations of the study has been added just before conclusions.

Minor comments:

1. The information has been added.

2. A new figure has been added in section 4.3 with EAE induction and experimental design scheme (Figure 3).

English:

We have revised typographical and grammatical mistakes.

Reviewer 2 Report

Comments and Suggestions for Authors

The authors provide justification for the approach of targeting IL-6 in the EAE model of MS and also specifically in old mice.  The study was well-designed with blinded rating of the signs and data analysis. The comparison of anti-IL6 ab administered i.p. after signs of disease first developed, did not ameliorate the clinical signs or lesions. However, it did increase the peripheral immunosuppressive response as measured by increases in immune check point inhibitors PD-1, TIM-3 in CD4 T cells and LAG-3 and TIM-3 in CD8 T cells.  This last detailed point should be included in the abstract. It is important to document findings of this nature ie abs  to IL-6 do not affect disease outcome and so this is a worthy manuscript

Author Response

We very much appreciate the constructive feedback on our manuscript. We also consider that documenting “negative” data is useful for the whole scientific community.

We have added the required information in the abstract: “However, IL-6 inhibition was associated with an increase in the peripheral immunosuppressive response: higher frequency of CD4+ T cells producing IL-10, and increased frequency of inhibitory immune check points PD-1 and Tim-3 on CD4 T cells and Lag-3 and Tim-3 on CD8+ T cells”.

Round 2

Reviewer 1 Report

Comments and Suggestions for Authors

The authors have addressed all comments. 

Comments on the Quality of English Language

A minor correction is needed in the revised text.